# Cervical length distribution among Brazilian pregnant population and risk factors for short cervix: A multicenter cross-sectional study

Kaline Gomes Ferrari Marquart[1], Thais Valeria Silva[1,2], Ben W. Mol[3], José Guilherme Cecatti[1], Renato Passini, Jr.[1], Cynara M. Pereira[1], Thaísa B. Guedes[1], Tatiana F. Fanton[1], Rodolfo C. Pacagnella[1]*, The P5 working group[¶]

1 Department of Obstetrics and Gynecology, School of Medicine, University of Campinas, Campinas, São Paulo, Brazil, 2 CISAM Maternity Hospital, University of Pernambuco, Recife, Pernambuco, Brazil, 3 Department of Obstetrics and Gynaecology, Monash University, Clayton, Victoria, Australia

¶ The Complete Membership of the Author Group can be found in the Acknowledgments
* rodolfop@unicamp.br

**Data Availability Statement:** The database is available at https://doi.org/10.25824/redu/P8PUWR.

## Abstract

### Objective

Since there are populational differences and risk factors that influence the cervical length, the aim of the study was to construct a populational curve with measurements of the uterine cervix of pregnant women in the second trimester of pregnancy and to evaluate which variables were related to cervical length (CL) ≤25 mm.

### Materials and methods

This was a multicenter cross-sectional study performed at 17 hospitals in several regions of Brazil. From 2015 to 2019, transvaginal ultrasound scan was performed in women with singleton pregnancies at 18 0/7 to 22 6/7 weeks of gestation to measure the CL. We analyzed CL regarding its distribution and the risk factors for CL ≤25 mm using logistic regression.

### Results

The percentage of CL ≤ 25mm was 6.67%. Shorter cervices, when measured using both straight and curve techniques, showed similar results: range 21.0–25.0 mm in straight versus 22.6–26.0 mm in curve measurement for the 5th percentile. However, the difference between the two techniques became more pronounced after the 75th percentile (range 41.0–42.0 mm straight x 43.6–45.0 mm in curve measurement). The risk factors identified for short cervix were low body mass index (BMI) (OR: 1.81 CI: 1.16–2.82), higher education (OR: 1.39 CI: 1.10–1.75) and personal history ([one prior miscarriage OR: 1.41 CI: 1.11–1.78 and ≥2 prior miscarriages OR: 1.67 CI: 1.24–2.25], preterm birth [OR: 1.70 CI: 1.12–2.59], previous low birth weight <2500 g [OR: 1.70 CI: 1.15–2.50], cervical surgery [OR: 4.33 CI: 2.58–7.27]). By contrast, obesity (OR: 0.64 CI: 0.51–0.82), living with a partner (OR: 0.76 CI: 0.61–0.95) and previous pregnancy (OR: 0.46 CI: 0.37–0.57) decreased the risk of short cervix.

**Funding:** This work was supported by the Bill & Melinda Gates Foundation [OPP1107597]. Under the grant conditions of the Foundation, a Creative Commons Attribution 4.0 Generic License has already been assigned to the Author Accepted Manuscript version that might arise from this submission. This work was supported by The Brazilian Ministry of Health, and the Brazilian National Council for Scientific and Technological Development (CNPq) [401615/20138]. The funders had no role in the design, development of the study, analysis, interpretation of data, writing the manuscript and in the decision to submit the article for publication.

**Competing interests:** The authors have declared that no competing interests exist.

## Conclusions

The CL distribution showed a relatively low percentage of cervix ≤25 mm. There may be populational differences in the CL distribution and this as well as the risk factors for short CL need to be considered when adopting a screening strategy for short cervix.

## Introduction

Transvaginal ultrasound (TVU) is the gold standard method of assessing cervical length in pregnant women, established by drawing a straight line between the internal and external orifice of the cervix; it provides objective and reproducible measurements [1–4]. TVU can also help to prevent prematurity because cervical length is one of the best predictors of preterm birth (PTB), and short cervical length may trigger interventions. Progesterone has a role in reducing spontaneous preterm in singleton pregnancies with cervical length (CL) ≤ 25 mm [5].

Although randomized studies have demonstrated benefits for the treatment of women with short cervix with progesterone in the reduction of PTB and consequently prevention of neonatal morbidity and mortality [6], the cutoff point defining short cervix that justifies interventions remains a matter of debate. The recommended cutoff point for intervention varied from 10 to 30 mm, with 25 mm being the most accepted cervical length that would trigger intervention [2, 7–9], including the recommendation of the American College of Obstetricians and Gynecologists (ACOG) [10].

The standard technique for measuring the cervix using TVU is to draw a straight line between the internal and external os [1–4]. Previous studies have already compared the straight technique with the curved technique, as well as the contribution of the volume of the uterine cervix for the diagnosis of short cervix. No technique showed better results compared to the standard technique [3, 4, 6].

There are populational differences in terms of the genesis of preterm labor and evidence associating cervical structure (length and dilation) with race and other social factors [11]. A retrospective cohort of singleton gestations without prior PTB undergoing universal second trimester ultrasound screening found that African-American women had a 2.8–fold increased risk of cervical length ≤25 mm compared to non-Hispanic white women in a low-risk population [12]. Another study of a prospective cohort of 5092 low risk women with singleton pregnancies who underwent TVU showed a relationship between mid-trimester cervical length and BMI, maternal age, maternal ethnicity, and parity [13]. These findings suggest that different groups of women may present specific characteristics and, therefore, it is necessary to identify conditions that may influence the cervical length and its ability to predict preterm labor.

The use of distribution curves of cervical length from generic population without consider populational differences can leading to unnecessary treatments or inaccurate risk estimation. Conversely, underestimating the risk for short cervix, may lead to failure to intervene. Nevertheless, few studies have been devoted to construction of specific population curves. Therefore, the aim of the present study was to describe curves for cervical length in singleton pregnant Brazilian women and to assess the risk factors associated with CL ≤25 mm.

## Materials and methods

This is a cross-sectional study from women including in the screening phase of the P5 Trial (Pessary Plus Progesterone to Prevent Preterm Birth Study). The P5 Trial was a randomized

controlled trial that compared the effectiveness of vaginal progesterone alone versus progesterone plus cervical pessary in women with short cervix, coordinated by the University of Campinas (Trial registration RBR-3t8prz) and approved by the Brazilian National Review Board (CONEP)—number 1.055.555 [14]. In July 2015, a TVU screening program was implemented in 17 institutions (nine states in three regions: South, Southeast and Northeast of Brazil) for 44 months period as the standard of care during routine second trimester ultrasonographic examination. In the current analysis, we studied 8167 singleton pregnant women using an online database from the screening phase of the P5 Trial.

All pregnant women attending the ultrasound department of these facilities at gestational age between 18 0/7 to 22 6/7 weeks of gestation were invited to participate. Before the exam, the women received information about the technique of ultrasound and about the P5 Study; all provided written informed consent. Women with painful contractions, vaginal bleeding, cerclage during current pregnancy before the screening, ruptured membranes diagnosed before screening, severe liver disease, cholestasis during this pregnancy, previous or current thromboembolism, placenta previa, cervical dilation greater than 1 cm, monoamniotic twin pregnancy, higher order multiple pregnancies (triplets or higher), and major fetal malformation or at least one fetus and stillbirth were not eligible for the study. All twin gestations were excluded from our current analysis.

The gestational age was calculated using the date of the last menstrual period (LMP) and was confirmed by a first trimester ultrasound. When there were discrepancies $\geq 7$ days, the first trimester ultrasound was used to calculate gestational age. Sociodemographic information, personal and previous gestational history and information about the current pregnancy were collected. After data collection and second trimester gestational US, TVU was performed using a GE Logic C5® equipment or similar with a 5–9-MHz transvaginal probe.

All sonographers were trained in cervical measurement according to the Fetal Medicine Foundation training program [15] and an additional training regarding the volume measurement. After emptying the bladder, the participant was placed in the dorsal lithotomy position. The transvaginal ultrasound probe was introduced and directed toward the anterior fornix, avoiding exerting undue pressure on the cervix, which may artificially increase the length. A sagittal view of the cervix was obtained and the endocervical mucosa was used as a guide to the proper position of the internal os. Four strategies of uterine cervical measurements were used in our study: straight line measurement (SL) between the internal to the external os, used for the primary outcome (distribution); curved measurement (CM) with two straight measurements respecting the endocervical canal pathway between the internal and external os (Figs 1 and 2); anteroposterior measurement near the insertion of the uterine arteries, in the middle third of the cervix; and transverse measurement rotating the transducer 90 degrees to allow transverse visualization of the cervix. The volume of the cervix was calculated using the formula for the volume of a cylinder, $\pi R^2 h$, where R is half the transverse diameter of the cervix, and h is the length. The curved measurement and the measurements for calculating the volume were used only for comparison purposes with the standard straight measurement. The presence or absence of sludge and funneling were also evaluated. Funneling was present when the internal os opening was in the form of "Y," "U" or "V," with a width greater than 5 mm. The time required to complete the exam was approximately 10 minutes.

The calculation of the sample size considered a standard deviation (SD) of 4.0 mm, a type I error of 0.05 and type II error of 0.2. The number estimated to be necessary to adequately power the study was 1500 women for each gestational age between 18 to 22 weeks, totaling a minimum number of 7500 pregnant women.

For the descriptive analysis, mean and percentiles for each measurement were obtained. Distribution curves were presented in graphics. The odds ratio (OR) and 95% confidence

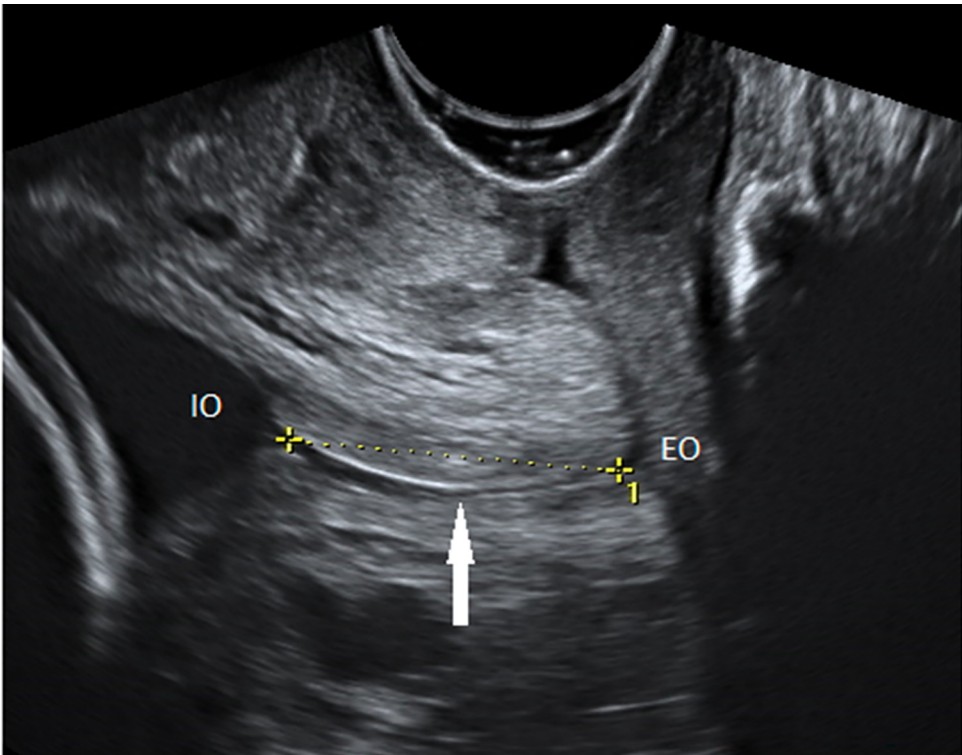

**Fig 1. CL measurement in straight line technique.** Transvaginal ultrasonography in sagittal section. The endocervical mucosa (arrow) is used as a guide to identify the internal (IO) and external (EO) os. The straight-line technique is presented (dashed line).

intervals (95% CI) for CL≤25mm were calculated. A stepwise multiple logistic regression analysis was used to select the variables to identify risk factors for short cervix. The following variables were used to estimate the model: maternal age (≤ 19, 20 to 34 and ≥ 35 years), schooling (until middle school and beyond high school), body mass index (BMI: low weight <18.5, normal weight 18.6 to 24.9, overweight 25 to 29.9 and obese ≥30), history of PTB and PTB <28 weeks, previous low birth weight (< 2500 g), cerclage in previous pregnancy, previous cervix surgeries, Mullerian malformations, non-spontaneous conception, marital status, number of births and miscarriage and the region of Brazil. Statistical analysis was performed using R software from the R Project for Statistical Computing (version 4.1.2).

## Results

A total of 7,844 of the 8,167 eligible pregnant women were included in the analysis. We excluded 323 participants: 48 due to lack of information and 275 twin pregnancies (Fig 3).

In our sample, almost 70% of women were between 20 and 34 years old, a total of 61.8% were overweight or obese, a quarter studied until middle school, 62.5% were non-white and 82.8% lived with their partner predominantly in the south and southeastern region. About obstetric history, 63% had previous pregnancies and 55% had previous births, 10.7% had PTBs, 3.5% had previous PTB <28 weeks and 9.1% with birth weight <2500 g. Regarding delivery, 2801 (35.7%) women had ≥ 1 previous vaginal deliveries and 2020 (25.7%) had previous C-sections; 0.4% had non-spontaneous conception, 0.4% had a previous cerclage, 1.3% had previous cervix surgeries, 1.5% presented uterine malformations, 1.3% women reported

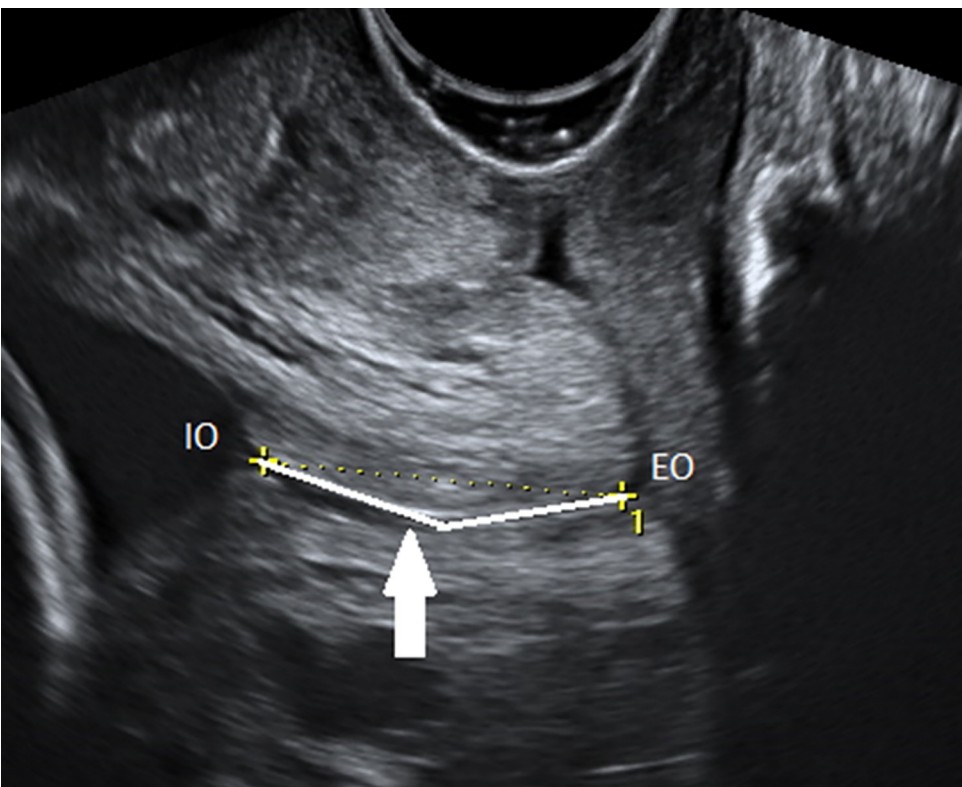

**Fig 2. CL measurement curve line technique.** Transvaginal ultrasonography in sagittal section. The endocervical mucosa (arrow) is used as a guide to identify the internal (IO) and external (EO) os. The curve technique is presented (continue line): two lines are drawn respecting the curvature of the endocervical canal.

active bleeding until the second trimester, and 4.6% and 3.0% presented sludge and funneling in TVU assessment, respectively (Table 1).

The mean cervical length in linear distance of our population was 36.9 mm, range 36.3 to 37.0 mm; in curve measurement, the mean was 40.1 mm range 38.2 to 39.6 mm (Table 2). In

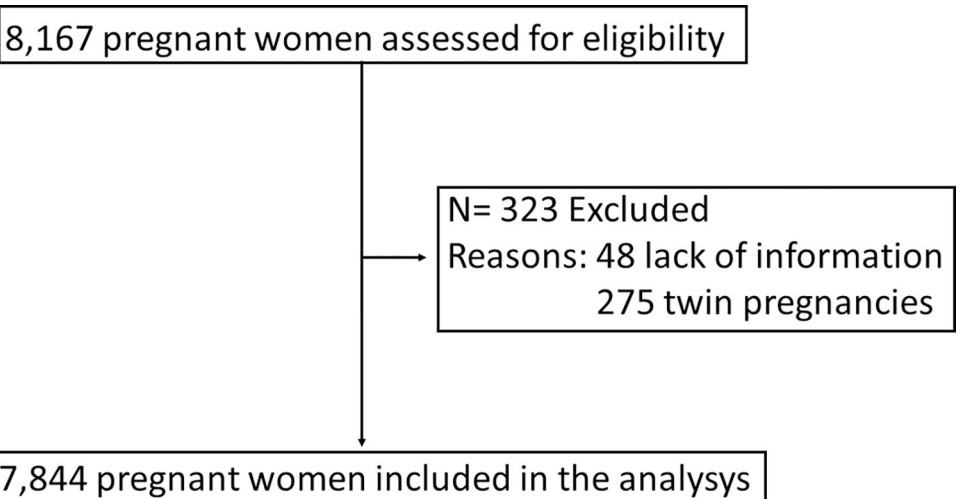

**Fig 3. Inclusion and exclusion flowchart.** Eligible pregnant women, excluded and included in the analysis.

**Table 1. Main sociodemographic characteristics of the sample population, mean, median of the ultrasonographic measurements in straight line of the uterine cervix in millimeters and percentage of cervical length ≤ 25 mm.**

| Characteristics | n (%) | Mean | Median | CL ≤ 25 mm (%) | p- value |
|---|---|---|---|---|---|
| **Maternal age** | | | | | 0.0019 |
| ≤ 19 | 924 (11.8) | 34 .99 | 35.00 | 9.09 | |
| 20 to 34 | 5425 (69.2) | 36.80 | 36.60 | 6.60 | |
| ≥ 35 | 1495 (19.0) | 38.44 | 38.00 | 5.42 | |
| **Schooling** | | | | | 0.0043 |
| Preschool, Elementary and Middle School | 1889 (24.1) | 37.33 | 37.00 | 5.24 | |
| High School and Higher education | 5955 (75.9) | 36.77 | 36.60 | 7.12 | |
| **Marital status** | | | | | 0.0018 |
| Without partner | 1351 (17.2) | 36.20 | 36.00 | 8.59 | |
| With partner | 6493 (82.8) | 37.05 | 37.00 | 6.27 | |
| **Region** | | | | | 0.0285 |
| Northeast | 2900 (37.0) | 37.03 | 37.00 | 5.86 | |
| South, southeast | 4944 (63.0) | 36.82 | 36.90 | 7.14 | |
| Race | | | | | 0.7630 |
| **White** | 2943 (37.5) | 37.10 | 37.00 | 6.56 | |
| **Non-white** | 4901 (62.5) | 36.78 | 36.50 | 6.73 | |
| **BMI (kg/m$^2$)** | | | | | <0.0001 |
| <18.5 | 178 (2.3) | 33.97 | 33.50 | 15.17 | |
| 18.6–24.9 | 2814 (35.9) | 35.84 | 35.70 | 8.17 | |
| 25–29.9 | 2630 (33.5) | 37.12 | 37.00 | 5.93 | |
| ≥ 30 | 2222 (28.3) | 38.22 | 38.00 | 4.95 | |
| **Numbers of pregnancies** | | | | | 0.0804 |
| 0 | 2900 (37.0) | 35.89 | 35.70 | 7.31 | |
| ≥ 1 | 4944 (63.0) | 37.49 | 37.20 | 6.29 | |
| **Numbers of births** | | | | | <0.0001 |
| 0 | 3528 (45.0) | 35.71 | 35.50 | 8.13 | |
| ≥ 1 | 4316 (55.0) | 37.88 | 38.00 | 5.47 | |
| **Numbers of vaginal births** | | | | | 0.62 |
| 0 | 5043 (64.3) | 36.64 | 36.20 | 6.56 | |
| ≥ 1 | 2801 (35.7) | 37.38 | 37.00 | 6.85 | |
| **Number of C-section** | | | | | <0.0001 |
| 0 | 5824 (74.3) | 36.27 | 36.00 | 7.73 | |
| ≥ 1 | 2020 (25.7) | 38.72 | 38.10 | 3.61 | |
| **Number of miscarriage** | | | | | 0.0025 |
| 0 | 5822 (74.2) | 37.06 | 37.00 | 6.15 | |
| 1 | 1369 (17.5) | 36.81 | 37.00 | 7.60 | |
| ≥ 2 | 653 (8.3) | 35.69 | 36.00 | 9.34 | |
| **History of preterm birth** | | | | | <0.0001 |
| Yes | 840 (10.7) | 35.41 | 36.00 | 13.10 | |
| No | 7004 (89.3) | 37.08 | 37.00 | 5.90 | |
| **History of preterm birth < 28 weeks** | | | | | <0.0001 |
| Yes | 272 (3.5) | 32.39 | 33.80 | 22.43 | |
| No | 7572 (96.5) | 37.06 | 37.00 | 6.10 | |
| **History birth weight (< 2500g)** | | | | | <0.0001 |
| Yes | 717 (9.1) | 35.28 | 35.70 | 13.95 | |
| No | 7127 (90.9) | 37.06 | 37.00 | 5.94 | |

*(Continued)*

**Table 1.** (Continued)

| Characteristics | n (%) | Mean | Median | CL ≤ 25 mm (%) | p- value |
|---|---|---|---|---|---|
| **Cerclage in previous pregnancy** | | | | | 0.0073 |
| Yes | 35 (0.4) | 32.09 | 33.00 | 20.00 | |
| No | 7809 (99.6) | 36.92 | 36.90 | 6.61 | |
| **Previous cervix surgeries** | | | | | <0.0001 |
| Yes | 102 (1.3) | 34.06 | 34.65 | 19.61 | |
| No | 7742 (98.7) | 36.94 | 37.00 | 6.50 | |
| **Uterine malformations** | | | | | 0.941 |
| Yes | 117 (1.5) | 37.51 | 38.00 | 6.84 | |
| No | 7727 (98.5) | 36.89 | 36.90 | 6.66 | |
| **Non-spontaneous conception** | | | | | 0.2768 |
| Yes | 33 (0.4) | 31.80 | 33.00 | 12.12 | |
| No | 7811 (99.6) | 36.92 | 37.00 | 6.64 | |
| **Sludge** | | | | | <0.0001 |
| Yes | 347 (4.6) | 29.00 | 30.50 | 30.55 | |
| No | 7497 (95.4) | 37.27 | 37.00 | 5.56 | |
| **Cervical Funneling** | | | | | <0.0001 |
| Yes | 229 (3.0) | 19.40 | 20.00 | 79.04 | |
| No | 7615 (97.0) | 37.48 | 37.00 | 4.49 | |
| n = 7844 | | | | | |

the descriptive analysis, a reduction in the CL from the twenty-first week of pregnancy, regardless of the technique used for measurement (straight or curve) can be observed (Figs 4 and 5). All pregnant women with SL CL measurement ≤25 mm was above the 5th percentile (Table 2 and Fig 4). Comparing the graphs for the straight and curve cervical length measurement grouped at the 5th percentile there was only a small amount of variation. However, in larger cervices, we observed a broader difference between the straight and curve measurement of the

**Table 2. Values of percentile 5, 10, 25, 50, 75, 90 e 95 for the cervical length measurement in linear distance between internal and external os and in curve by ultrasonography according to gestational age.**

| Gestational age (weeks) | | Cervical length in linear distance (mm) | | | | | | | |
|---|---|---|---|---|---|---|---|---|---|
| | Mean | p 5 | p 10 | p 25 | p 50 | p 75 | p 90 | p 95 | CL ≤ 25 mm (%) |
| 18 | 37.0 | 24.4 | 28.3 | 32.0 | 37.0 | 41.5 | 47.0 | 51.0 | 5.7 |
| 19 | 36.9 | 25.0 | 28.4 | 32.3 | 36.3 | 41.0 | 46.0 | 49.0 | 5.3 |
| 20 | 37.2 | 25.0 | 28.7 | 32.8 | 37.0 | 41.5 | 47.0 | 50.4 | 5.4 |
| 21 | 37.2 | 24.0 | 28.0 | 33.0 | 37.0 | 42.0 | 47.0 | 51.0 | 6.4 |
| 22 | 36.2 | 21.0 | 25.6 | 32.0 | 36.3 | 41.0 | 46.3 | 50.0 | 9.6 |
| Total | 36.9 | 23.8 | 27.8 | 32.3 | 36.9 | 41.6 | 47.0 | 50.0 | 6.7 |
| n = 7844 | | | | | | | | | |
| Gestational age (weeks) | | Cervical length in curve (mm) | | | | | | | |
| | Mean | p 5 | p 10 | p 25 | p 50 | p 75 | p 90 | p 95 | CL ≤ 25 mm* (%) |
| 18 | 41.8 | 26.0 | 30.0 | 34.6 | 39.6 | 45.0 | 54.0 | 66.0 | 5.1 |
| 19 | 40.3 | 25.9 | 30.4 | 34.0 | 39.0 | 44.5 | 52.4 | 63.0 | 4.9 |
| 20 | 40.5 | 26.0 | 30.7 | 35.0 | 39.0 | 45.0 | 53.0 | 60,2 | 4.5 |
| 21 | 40.1 | 25.1 | 29.5 | 34.3 | 39.0 | 45.0 | 52.0 | 58.5 | 5.6 |
| 22 | 38.5 | 22.6 | 27.0 | 33.6 | 38.2 | 43.6 | 49.5 | 55.0 | 8.3 |
| Total | 40.1 | 25.0 | 29.1 | 34.0 | 39.0 | 44.6 | 52.1 | 59.0 | 6.1 |
| n = 7765 | *Considering the curved technic | | | | | | | | |

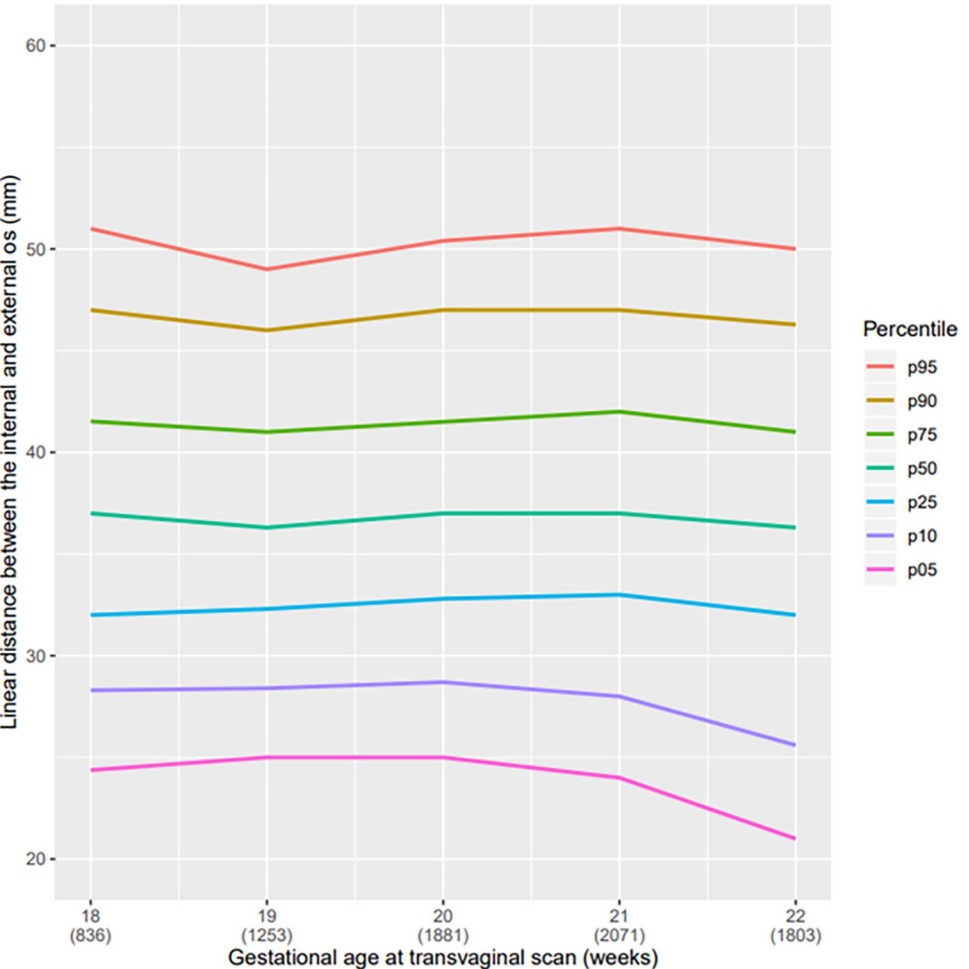

**Fig 4. Curve of percentile values for the linear CL measurement.** Curve of percentile values for the linear distance between the internal and external os according to gestational age (weeks) at transvaginal scan.

cervix. The median CL at 20 weeks of GA was 2 mm higher using the CM than using the SL; in the 95th percentile, this difference was almost 10 mm (Table 2). We observed that the cervical volume slightly increased with progression of gestational age (Fig 6).

The lowest mean cervix lengths were observed in women with cervical funneling (19.40 mm) and sludge (29.0 mm) followed by those who had non-spontaneous conception (31.80 mm), previous history of cerclage (32.09 mm), preterm birth <28 weeks (32.39 mm) and in low-weight women (33.97 mm). Of 7844 women, 523 (6.67%) had CL ≤ 25 mm. The percentage of CL ≤25 mm was high among women with cervical funneling, sludge, and other clinical condition related to preterm birth were higher than in the total sample (Table 1).

Considering 25 mm as a cutoff point for risk of preterm birth, we sought to identify variables associated with it. The variables significantly associated with CL ≤ 25 mm were as follows: BMI ≤ 18.5 (aOR: 1.81 CI: 1.16–2.82), higher levels of education (aOR: 1.39 CI: 1.10–1.75), one or more miscarriages (respectively aOR: 1.41 CI: 1.11–1.78 and aOR: 1.67 CI: 1.24–2.25), previous history of preterm birth < 28 weeks (aOR: 2.72 CI: 1.79–4.15), preterm birth (aOR: 1.70 CI: 1.12–2.59), previous child with low birth weight < 2500 g (aOR: 1.70 CI: 1.15–2.50) and history of cervix surgery (aOR: 4.33 CI: 2.58–7.27). By contrast, characteristics inversely associated to CL ≤ 25 mm were living with a partner (aOR: 0.76 CI: 0.61–0.95),

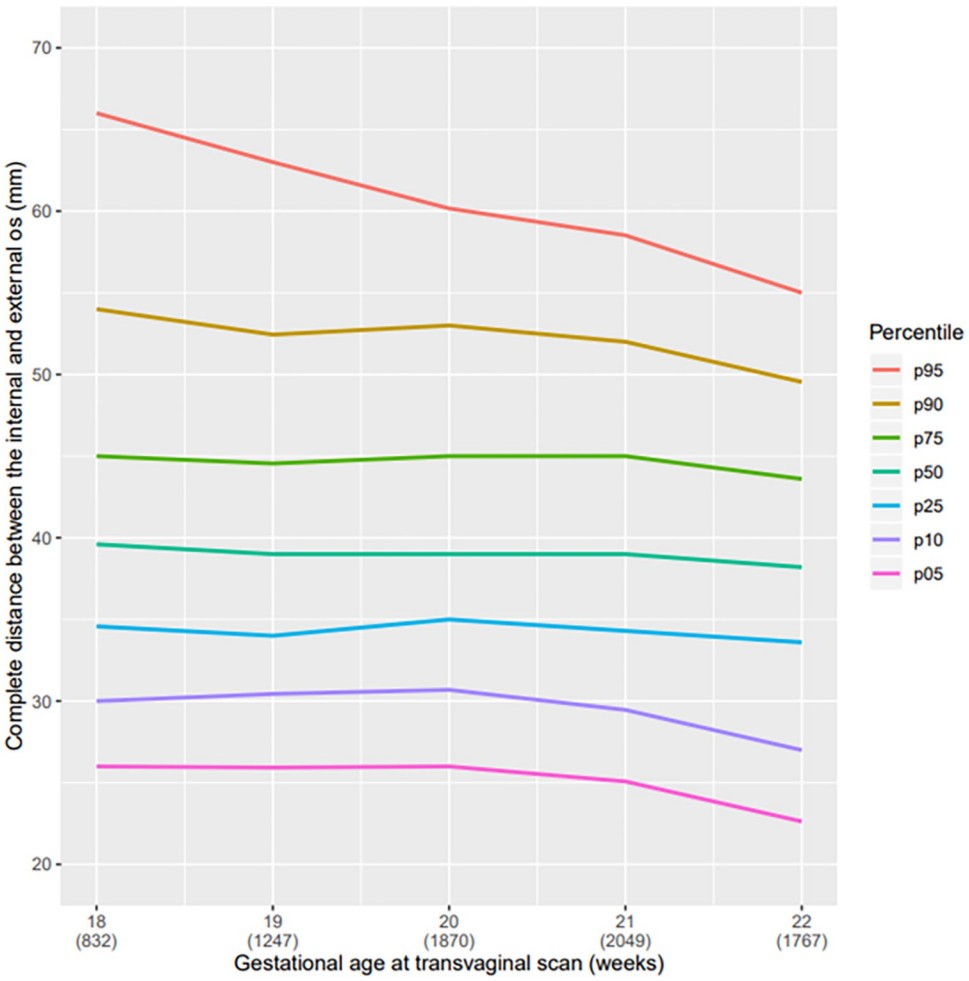

**Fig 5. Curve of percentile values for the curve CL measurement.** Curve of percentile values for the curve distance between the internal and external os according to gestational age (weeks) at transvaginal scan.

maternal overweight (aOR: 0.74 CI: 0.60–0.92), obesity (aOR: 0.64 CI: 0.51–0.82) and at least one previous delivery (aOR: 0.46 CI: 0.37–0.57) (Table 3).

We also assessed women without previous pregnancies separately from those who had at least one previous pregnancy. Women with previous pregnancies and with previous deliveries had reduced risk of CL ≤ 25 mm (aOR: 0.30 CI: 0.22–0.41). Moreover, those who had a history of PTB birth < 28 weeks had 2.7-fold increased risk for CL ≤ 25 mm (aOR: 2.77 CI: 1.82–4.22) as well as women who had a previous child with low birthweight <2500 g (aOR: 1.74 CI: 1.17–2.57). In the group of women without previous pregnancies, those living with their partners had a lower frequency of CL ≤ 25 mm (aOR: 0.68 CI: 0.50–0.91) and living in the southeast and south regions were associated to a CL ≤ 25 mm (aOR: 1.41 CI: 1.04–1.90). In both groups, previous cervix surgery significantly increased the risk of CL ≤ 25 mm (multiparous: aOR: 4.54 CI: 2.43–8.47 and nulliparous: aOR: 3.77 CI: 1.48–9.60) (Tables 4 and 5).

## Discussion

We determined the CL distribution among second trimester Brazilian pregnant women. The distribution showed a low percentage of CL ≤25 mm. The risk factors associated with

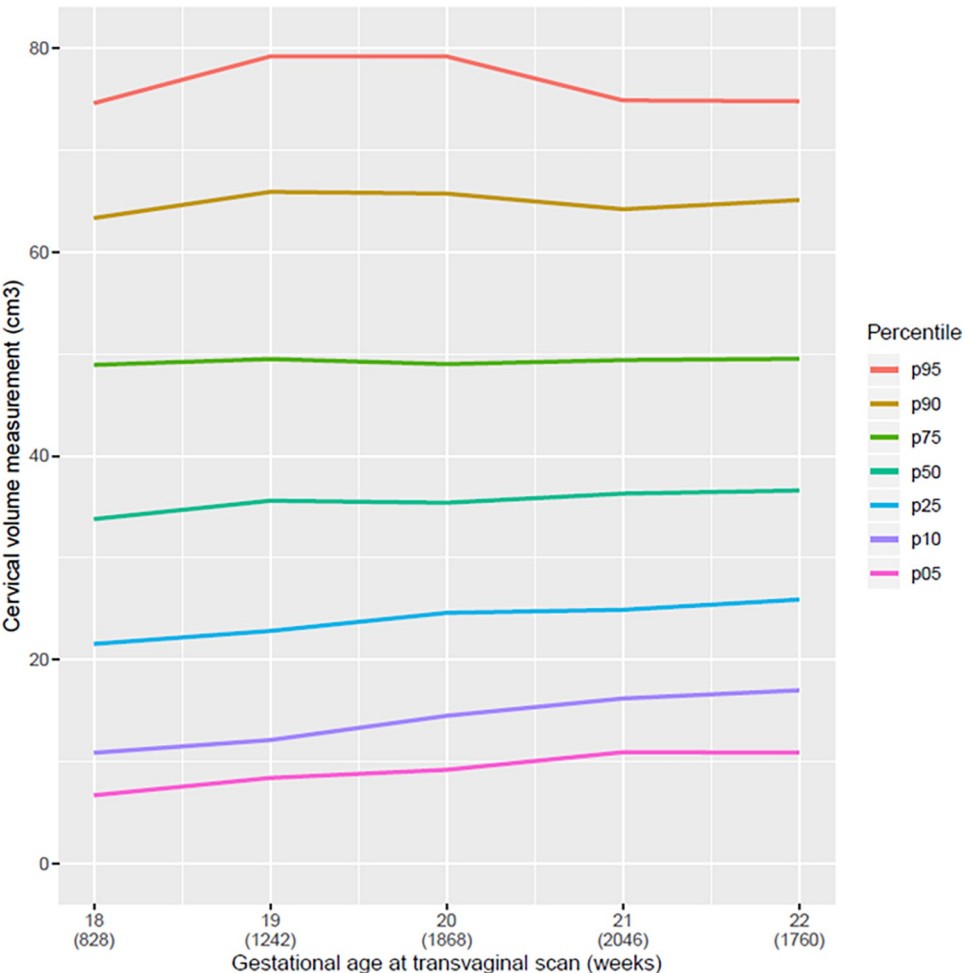

**Fig 6. Curve of percentile values for the volume of the uterine cervix according to gestational age (weeks) at transvaginal scan.** Curve of percentile values for the volume of the uterine cervix according to gestational age (weeks) at transvaginal scan.

increased risk for CL ≤25 mm were as follows: low BMI, high levels of education, previous miscarriage, prior PTB (especially if <28 weeks), previous low birthweight <2500 g and prior cervical surgery.

Iams et al. were among the pioneers in proposing reference values for CL. For women at 22-week's gestation, we found very similar measurements for the 5[th], 10[th] and 25[th] percentiles [2]. Our 5[th] and 10[th] percentiles, however, were similar to those previously proposed. Nevertheless, even considering women with a CL ≤25 mm as having an increased risk for preterm birth, our data corroborates the fact that there may be populational differences in the CL distribution and its relation to preterm birth risk compared to the literature [2, 16, 19].

Studies have proposed cervical distributions curves for the Brazilian population considering population characteristics. In general, the 50[th] and 95[th] percentiles are similar to those of our study; however, for the lower percentiles, we obtained slightly different values than others for the lower percentiles [16, 17].

Even within a single country, it is also necessary to be aware of the importance of intra-population differences. A prospective cohort found that Afro-Caribbean women had a shorter cervix than did Caucasian women [18]. Similar findings were identified in a retrospective cohort

**Table 3. Multiple analysis for cervical length $\leq$ 25 mm.**

| Variables | p-value | aOR (95% CI) |
|---|---|---|
| **Marital status (living with partner)** | 0,018 | 0.76 (0.61–0.95) |
| **BMI (kg/m$^2$)** | | |
| <18.5 | 0,009 | 1.81 (1.16–2.82) |
| 25–29.9 | 0,007 | 0.74 (0.60–0.92) |
| $\geq$ 30 | 0.000 | 0.64 (0.51–0.82) |
| **Schooling (High School and Higher education)** | <0.0001 | 1.39 (1.10–1.75) |
| **Numbers of previous births $\geq$ 1** | <0.0001 | 0.46 (0.37–0.57) |
| **Numbers of miscarriages** | | |
| 1 | 0,005 | 1.41 (1.11–1.78) |
| $\geq$ 2 | 0,001 | 1.67 (1.24–2.25) |
| **Previous history of preterm birth < 28 weeks $\geq$ 1** | 0.000 | 2.72 (1.79–4.15) |
| **Previous history of preterm birth** | 0,013 | 1.70 (1.12–2.59) |
| **Previous birth weight (< 2500g)** | 0,008 | 1.70 (1.15–2.50) |
| **Previous cervix surgeries** | <0.0001 | 4.33 (2.58–7.27) |
| n = 7844 | | |

conducted in the US involving 16,598 women in the second trimester of pregnancy, suggesting that a short cervix definition should differ between ethnic groups within the same population [19].

In 2020, a prospective Asian cohort study involving 1013 women found significant difference between the mean cervical measurement by population group (Chinese 32.2 ± 0.77 mm, Malay 31.3 ± 0.69 mm, Indian 29.7 ± 0.70, Others 33.3 ± 0.82 mm) [20]. Our study, thus, reinforce that a single distribution curve for cervical measurement, without considering the different population characteristics, may not represent all women equally and could inefficiently guide preventive measures for prematurity.

With respect to different techniques to measure the cervix, shorter cervixes when measured by both straight and curve techniques, do not differ substantially. In both techniques, there is a reduction of CL as the pregnancy advances [21, 22]. We observed that CL reduces more significantly after 21 weeks, regardless of the technique used for assessment and this pattern is reported in literature. However, in longer cervix, we observed that the difference between the two techniques becomes more pronounced.

**Table 4. Multiple analysis for cervical length $\leq$ 25 mm in women with previous pregnancies.**

| Variables | p-value | aOR (95% CI) |
|---|---|---|
| **BMI (kg/m$^2$)** | | |
| <18.5 | 0.292 | 2.10 (1.08–4.10) |
| 25–29.9 | 0.117 | 0.80 (0.61–1.06) |
| $\geq$ 30 | 0.001 | 0.59 (0.43–0.81) |
| **Schooling (High School and Higher education)** | 0.023 | 1.40 (1.05–1.87) |
| **Numbers of previous births $\geq$ 1** | <0.0001 | 0.30 (0.22–0.41) |
| **Previous history of preterm birth < 28 weeks $\geq$ 1** | <0.0001 | 2.77 (1.82–4.22) |
| **Previous history of preterm birth** | 0,014 | 1.69 (1.11–2.58) |
| **Previous birth weight (< 2500g)** | 0,006 | 1.74 (1.17–2.57) |
| **Previous cervix surgeries** | <0.0001 | 4.54 (2.43–8.47) |
| n = 4944 | | |

**Table 5. Multiple analysis for cervical length ≤ 25 mm in women without previous pregnancies.**

| Variables | p-value | aOR (95% CI) |
|---|---|---|
| **Marital status (living with partner)** | 0,011 | 0.68 (0.50–0.91) |
| **BMI (kg/m$^2$)** | | |
| <18.5 | 0,130 | 1.59 (0.87–2.89) |
| 25–29.9 | 0,013 | 0.63 (0.44–0.91) |
| ≥ 30 | 0,132 | 0.75 (0.51–1.09) |
| **Region (South, southeast)** | 0,027 | 1.41 (1.04–1.90) |
| **Previous cervix surgeries** | 0,005 | 3.77 (1.48–9.60) |
| N = 2900 | | |

This underestimation in the values of the last quartile changes the design of the distribution curve. Considering the importance of building reference curves that respect the cervical anatomy, we suggest that the most adequate method to measure the cervix is respecting the curvature, however, for short cervix, which is the main factor related to spontaneous PTB, the straight-line measurement for cervical length may be the best strategy. Previous studies have already compared the straight technique with the curved technique [23], as well as the contribution of the volume of the uterine cervix for the diagnosis of short cervix. No technique showed better results compared to the standard technique [24–29].

We also observed that the volume of the cervix increased slightly over the course of gestation despite the progressive shortening of the longitudinal measurement of the cervix. In other words, the cervix becomes shorter but wider [23]. Although many studies have shown a correlation between cervical volume and the ability of this measure to contribute to the prediction of the risk of prematurity, none has demonstrated additional benefits in relation to the longitudinal cervical measurement technique [24–28].

Regarding the risk factors for short cervix, level of schooling is a social aspect that is related to health improvement; however, it was found that extremely high maternal education did not confer more protection against PTB [29]. In high-income countries, a higher level of education is also associated with increasing working day for women. High-level education provokes an overloaded of responsibilities and stress, including employment relationships, excessive time into the traffic, less time to the physiological needs (like time to sleep, rest, and healthy nutrition), and less time for family care and domestic tasks [28].

We found that CL was shorter in pregnant women ≤19 years old than those >20 years old. In the literature, young pregnant women are at increased risk for spontaneous PTB [30], which may be due to biological immaturity of the female genital tract [31, 32], social and behavioral factors [18], and intra-amniotic infections as a consequence of genital tract infections [31].

We as well found that low BMI was associated with CL≤ 25 mm. This result confirms the findings from other studies showing the same relationship between shorter CL and lower BMI [13, 18, 33]. There appears to be a correlation between low pre-pregnancy BMI and low weight gain during pregnancy with spontaneous preterm birth [34, 35]. On the other hand, we found a lower frequency of CL ≤25 mm in obese than in underweight women. A systematic review showed that, compared to normal weight women, pre-obese women and those with grade I obesity had a 15% reduction in their risk of spontaneous PTB and the prevalence of short cervix was significantly lower in obese compared to normal or underweight women [34]. By contrast, other studies showed relationships between obesity and prematurity [33–38], mostly related to therapeutic PTB [39]. A theory to explain this cervical behavior on pregnancies with low BMI is related to acquired collagen deficiency.

Among the strongest risk factors for PTB we found the previous history of spontaneous preterm delivery, particularly if it occurred early in pregnancy [40]. Our study reinforces this argument and shows an association between short cervix and history of PTB [37]. Nevertheless, in literature, nulliparity appears to increase the risk for PTB [30]. The mechanisms by which nulliparity can lead to PTB remain poorly understood. We noted a smaller mean and median of CL as well as a larger percentage of CL $\leq 25$ mm in nulliparous as compared to multiparous women.

The strength of this study is that we included a large sample of cervical measurements from Brazilian pregnant women with singletons in the second trimester, establishing reference values; therefore, external validation is possible. Limitations of our study include the cross-sectional design that prevented establishment of correlations between the two techniques for measuring the cervix (straight and curve) and the outcome (PTB). In addition, we analyzed data from 836 and 1253 pregnant women with gestational ages of 18 and 19 weeks, respectively, thereby failing to reach the calculated sample size at these gestational ages. The measurements were performed by different, albeit trained, professionals from different facilities, which might have included a sort of bias in measurements. We also do not have information regarding the outcomes for most women, which would have added information.

Considering the results of the multiple analysis, because the universal screening of pregnant women in the second trimester remains controversial and is not recommended by the main gynecology and obstetrics societies [41–43], as well as the fact that we recognize that women with CL $\leq 25$ mm due to population differences are at different risks for PTB, we can propose that risk factors for CL $\leq 25$mm in mid-trimester for Brazilian singleton pregnant women as follows: low BMI, high levels of education, previous miscarriage, prior PTB (especially if $<28$ weeks), previous low birth weight $<2500$ g and prior cervical surgery. However, as the prevalence of PTB in Brazil is high, in places where financial resources are available and easy access to transvaginal ultrasound, we recommend that universal screening in the second trimester of pregnancy should be implemented.

## Conclusion

The reference CL distribution curves should consider populational characteristics since physicians may use it as a strategy to prevent preterm birth in clinical practice. Doing so, it will enable a more efficient diagnosis of short cervix and its association with prematurity, allowing assertive medical decisions. Moreover, we suggest that subsequent studies should consider these populational characteristics to build new distribution curves and define specific screening strategies for different populations to prevent premature delivery.

## Supporting information

**S1 Appendix. P5 trial description.**
(DOCX)

**S2 Appendix. STROBE statement checklist.**
(DOCX)

**S3 Appendix. The P5 working group.**
(DOCX)

## Acknowledgments

### *P5 working group

Amanda Dantas; Anderson Borovac-Pinheiro; Antonio Fernandes Moron; Carlos Augusto Santos Menezes; Cláudio Sérgio Medeiros Paiva; Cristhiane B Marques; Cynara Maria Pereira; Djacyr Magna Cabral Paiva; Elaine Christine Dantas Moisés; Enoch Quinderé Sá Barreto; Felipe Soares; Fernando Maia Peixoto-Filho; Francisco Edson de Lucena Feitosa; Francisco Herlanio Costa Carvalho; Jessica Scremin Boechem; João Renato Benini-Jr.; Karayna Gil Fernandes; Kleber Cursino Andrade; Leila Katz; Maíra Rossmann Machado; Marcelo L Nomura; Marcelo Marques Souza Lima; Marcelo Santucci Franca; Marcos Nakamura-Pereira; Maria Julia Miele; Maria Laura Costa; Mário Dias Correia Jr; Nelson Sass; Renato T Souza; Rodrigo Pauperio Soares Camargo; Samira Maerrawi Haddad; Sérgio Martins-Costa; Silvana F Bento; Silvana Maria Quintana; Stéphanno Gomes Pereira Sarmento;

## Author Contributions

**Conceptualization:** Ben W. Mol, José Guilherme Cecatti, Rodolfo C. Pacagnella.

**Data curation:** Kaline Gomes Ferrari Marquart, Tatiana F. Fanton, Rodolfo C. Pacagnella.

**Formal analysis:** Kaline Gomes Ferrari Marquart, Thais Valeria Silva, Tatiana F. Fanton, Rodolfo C. Pacagnella.

**Funding acquisition:** Ben W. Mol, José Guilherme Cecatti, Rodolfo C. Pacagnella.

**Investigation:** Kaline Gomes Ferrari Marquart, Thais Valeria Silva, Renato Passini, Jr., Cynara M. Pereira, Thaísa B. Guedes.

**Methodology:** Kaline Gomes Ferrari Marquart, Thais Valeria Silva, Rodolfo C. Pacagnella.

**Resources:** Ben W. Mol, Cynara M. Pereira, Thaísa B. Guedes.

**Software:** Tatiana F. Fanton.

**Supervision:** Rodolfo C. Pacagnella.

**Validation:** José Guilherme Cecatti, Rodolfo C. Pacagnella.

**Writing – original draft:** Kaline Gomes Ferrari Marquart, Thais Valeria Silva.

**Writing – review & editing:** Ben W. Mol, José Guilherme Cecatti, Renato Passini, Jr., Rodolfo C. Pacagnella.

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
