## [Decision Letter · Decision Letter 0]

6 Oct 2021

PONE-D-21-13533Cervical length distribution among Brazilian pregnant population and risk factors for short cervix: a multicenter cross-sectional studyPLOS ONE

Dear Dr. Pacagnella,

Thank you for submitting your manuscript to PLOS ONE. After careful consideration, we feel that it has merit but does not fully meet PLOS ONE’s publication criteria as it currently stands. Therefore, we invite you to submit a revised version of the manuscript that addresses the points raised during the review process.

We look forward to receiving your revised manuscript.

Kind regards,

Federico Ferrari

Academic Editor

PLOS ONE

Journal Requirements:

4. One of the noted authors is a group or consortium "The P5 working group". In addition to naming the author group, please list the individual authors and affiliations within this group in the acknowledgments section of your manuscript. Please also indicate clearly a lead author for this group along with a contact email address.

Reviewers' comments:

Reviewer's Responses to Questions

**Comments to the Author**

1. Is the manuscript technically sound, and do the data support the conclusions?

Reviewer #1: Partly

Reviewer #2: Yes

2. Has the statistical analysis been performed appropriately and rigorously? 

Reviewer #1: Yes

Reviewer #2: Yes

3. Have the authors made all data underlying the findings in their manuscript fully available?

Reviewer #1: Yes

Reviewer #2: Yes

4. Is the manuscript presented in an intelligible fashion and written in standard English?

Reviewer #1: Yes

Reviewer #2: No

5. Review Comments to the Author

Reviewer #1: Abstract

A brief mention of why this topic is important would be helpful.

The mention of straight versus curved measurements does not appear to align with the rest of the abstract. This should be clearer.

Intro

Page 5 - Second paragraph - recommend mentioning ACOG's use of 25 mm cutoff

Page 5 - Paragraph 3 is well written but I would also include mention of the techniques for measuring CL. What does AIUM say? How did previous publications describe their approach to CL measurements?

Page 6 - First paragraph would use "distribution curves" or "population curves" as the wording is confusing

Materials/Methods

I believe this is a case-control study

Why were so many types of subjects excluded?

Page 7 Paragraph 3 - what is the reference for the training?

Results

Page 12 - The mention of cervical volume was not described in the intro and its utility/relevance was not described

Was an adjusted analysis done to assess for confounders? Were Tables 3/4/5 adjusted?

Table 5 isn't labeled in the text

Data points should be shown as bell curves as well and authors should consider overlapping these images

Some of the results are confusing regarding the difference in the two measurement techniques.

Discussion

Which values did the authors use the straight line or the curved for the discussion?

Comparison of previous studies evaluating cervical volume to their results would benefit this manuscript

Comparison of previous studies evaluating techniques for cervical length measurement would benefit this manuscript

Majority of discussion and conclusion are well written

Lastly, the authors should consider the write up of the methodology for cervical measurement (straight vs curved) as a separate paper and focus on use of cervical length and cervical volume, as the finer points of the paper get lost in the results and discussion due to the many analyses.

Reviewer #2: In this manuscript, Marquat and collaborators present results of the association between cervical length distribution and risk factors for short cervix. The study was conduct 8,167 pregnant women in Brazil. I can see interesting results in this study, this article has a major methodological flaw, mainly because it is limited and superficial. Thus, without some basic information, it is difficult to judge the quality of the study. In addition, the authors should clearly indicate several important limitations observed in this work and correct some errors before publication.

• Line 45-47: I don’t understand what are they saying, please work on it and more informative.

• Line 54-55: Didn’t show in paper those are not significant to your research.

• Line 76: Review these references to fit the journal guidelines and also Please check grammar.

• Line 78: Cannot use this type of word. Please review it.

• Line 109-186: In material and method section is not clear, rewrite again kindly, for data analysis which software are authors use?

• Line 159-167: Don’t use figure in methods and where is your statical analysis? and check grammar.

• Line 189-191: Please add these lines in “material and method” section

• Line 192: Don’t use figure, please rewrite this.

• Line 227-238: Didn’t need use any figure here.

• Line 245: Check Grammar

• Line 277: Check Grammar

• Line 296: Check the spelling

• Line 327 – 332: The first sentence is not clear. It seems incomplete. Rewrite the full lines.

• Line 378-383: Rewrite the conclusion again.

6. PLOS authors have the option to publish the peer review history of their article (what does this mean?). If published, this will include your full peer review and any attached files.

Reviewer #1: **Yes: **Rachel Harrison

Reviewer #2: No

---

## [Author Response · Author response to Decision Letter 0]

18 Nov 2021

Reviewer 1

Abstract

A brief mention of why this topic is important would be helpful.

Thank you for the suggestion. We believe that the importance of this study is in contributing to screening strategies since there are populational differences and risk factors that influence the cervical length (line 69-72). We considered this 

The mention of straight versus curved measurements does not appear to align with the rest of the abstract. This should be clearer.

As for the alignment of the straight and curved technique, we made changes to the text to make it clear that for statistical analysis (logistic regression), we used the straight-line technique, since it is the gold standard technique for performing measurements of the uterine cervix. We used the curved technique only to compare with the standard technique. We excluded this information from the abstract as this could bring confusion.

Intro

Page 5 - Second paragraph - recommend mentioning ACOG's use of 25 mm cutoff

Thank you very much. We acknowledge that ACOG’s uses 25 mm as the cutoff point for a short cervix and we included this reference: Prediction and Prevention of Spontaneous Preterm Birth: ACOG Practice Bulletin, Number 234. Obstet Gynecol. 2021;138(2):e65-e90.

Page 5 - Paragraph 3 is well written but I would also include mention of the techniques for measuring CL. What does AIUM say? How did previous publications describe their approach to CL measurements?

Thank you for the comment. We added a mention about the technique in the first paragraph to make it clearer that transvaginal ultrasound is the gold standard for measuring the cervix and that the straight-line technique is the established method. 

The AIUM also advises that transvaginal ultrasound is the best method for evaluating the cervix and describes the technique for measuring the cervix in straight line. Thus, we add the following reference: AIUM Practice Parameter for the Performance of Limited Obstetric Ultrasound Examinations by Advanced Clinical Providers. J Ultrasound Med. 2018;37(7):1587-96.

As already established, the straight technique is the standard measure [2-4, 10, 15-17]. However, previous studies [18, 23] have already shown that the straight measurement can underestimate the measurement of the cervix, since longer cervix tend to adopt a more curvilinear shape. The curved measurement, as it follows the anatomy of the cervix, gives us a real measure of the cervical length. Our study confirmed these findings. Our objective was to construct distribution curves for the uterine cervix of Brazilian pregnant women in the second trimester, and a multicenter study had never been carried out in Brazil. Therefore, we think it would be important to compare the technique that respects the anatomy (curved technique) with the standard technique. However, as the short cevix tends to assume a straightened anatomical shape, we observe that the straight technique remains an effective method of diagnosis.

Page 6 - First paragraph would use "distribution curves" or "population curves" as the wording is confusing

We changed to distribution 

Materials/Methods

I believe this is a case-control study

We understand the study design as a cross-sectional study. Although we identified short and no-short cervices within our population, we obtained data from all the participants (cervical measurement, sociodemographic data, and obstetric and gynecological history) regardless any characteristics of cervical length. We did not select our study population considering cervical length, we performed the group identification in the analysis only.

Why were so many types of subjects excluded?

Thank you for your question. We used data from a large randomized, multicenter study that aimed to compare the effectiveness of using progesterone versus progesterone and pessary in patients with a short cervix. Therefore, as our study is a secondary analysis of this large study, it was necessary to follow the exclusion criteria of the main study and, from this database, we only excluded twin pregnancies. 

Page 7 Paragraph 3 - what is the reference for the training?

We appreciate your suggestion. We added the reference: The Fetal Medicine Foudation. The FMF certification cervical assessment. Available in: https://fetalmedicine.org/fmf-certification-2/cervical-assessment-1. Accessed on November 2, 2021. 

Results

Page 12 - The mention of cervical volume was not described in the intro and its utility/relevance was not described

Thank you very much. We added one paragraph in the introduction: “The standard technique for measuring the cervix using TVU is to draw a straight line between the internal and external os [1-4]. Previous studies have already compared the straight technique with the curved technique [23], as well as the contribution of the volume of the uterine cervix for the diagnosis of short cervix. No technique showed better results compared to the standard technique [24-29]”. 

We also described in the discussion that volume of the cervix was not relevant for diagnosis of short cervix, and we observed that short cervix tend to be rectified so curve straight line could be the best strategy. 

Was an adjusted analysis done to assess for confounders? Were Tables 3/4/5 adjusted? 

Yes, an adjusted analysis was performed to assess confounding factors. Tables 3 and 4 are analyzes of pre-determined subgroups in the study design and worked statistically with multiple logistic regression to reduce confounding factors.

Table 5 isn't labeled in the text

There was no table 5, however, I agree that the way table 4 was structured was confusing. In this way, we broke down table 4 and created table 5. Thanks for the suggestion.

Data points should be shown as bell curves as well and authors should consider overlapping these images

We understand that presenting data as we present in the text may be more useful for clinicians to use the curves as presenting in a bell shape.

Some of the results are confusing regarding the difference in the two measurement techniques.

We tried to reduce the confusion changing some wording and the data in tables. 

Discussion

Which values did the authors use the straight line or the curved for the discussion?

Thank you very much for your observation, it was not clearly written. We used the straight technique for the analysis, which is the standardized technique for measuring the cervical length of the uterine cervix. We added this information in the first paragraph to make it clearer: “We determined the CL distribution (in straight line) among second trimester Brazilian pregnant women”.

Comparison of previous studies evaluating cervical volume to their results would benefit this manuscript.

We compared with other studies that also evaluated volume of the cervix, however, it has not been shown to be superior to diagnose short cervix compared to the standard method (straight technique): “We also observed that the volume of the cervix increased slightly over the course of gestation despite the progressive shortening of the longitudinal measurement of the cervix. In other words, the cervix becomes shorter but wider [23]. Although many studies have shown a correlation between cervical volume and the ability of this measure to contribute to the prediction of the risk of prematurity, none has demonstrated additional benefits in relation to the longitudinal cervical measurement technique [24-28].”

Comparison of previous studies evaluating techniques for cervical length measurement would benefit this manuscript

Thank you for your suggestion. We agreed and added comparison of previous study: “Similar results were found in a prospective Dutch cohort involving 508 women aged 18-22 weeks who identified greater differences between line and trace measurement techniques above P95 (line 51.1mm x trace 55mm, p <0.0001) [21]. This result is because the straight measurement underestimates the biometry of the cervix. The construction of the reference curve for the measurement of the cervix, considering the straight line as a reference, specially underestimates the values above 75th percentile when the difference between the straight and curve measurements assumes a tendency to be > 4mm, which according to the Fetal Medicine Foundation (FMF) protocol cannot be considered an operator-dependent difference [15].”

Majority of discussion and conclusion are well written

Thank you very much.

Reviewer 2

Line 45-47: I don’t understand what are they saying, please work on it and more informative.

Thank you for your suggestion. We tried to rewrite more clearly. 

" Since there are populational differences and risk factors that influence the cervical length, the aim of the study was to construct a populational curve with measurements of the uterine cervix of pregnant women in the second trimester of pregnancy and to evaluate which variables was related to cervical length ≤25 mm".

Line 54-55: Didn’t show in paper those are not significant to your research.

Thank you very much for your observation. 

"showed similar results: range 21.0–25.0 mm in straight versus 22.6– 26.0 mm in curve measurement for the 5th percentile".

Line 76: Review these references to fit the journal guidelines and also Please check grammar.

We adapted the references according to the journal guidelines.

"TVU can also help to prevent preterm birth (PTB) because cervical length is one of the best predictors of preterm birth, and short cervical length may trigger interventions"

Line 78: Cannot use this type of word. Please review it.

Thank you. We rephrased the sentence.

"The progesterone has a role in reducing spontaneous preterm in singleton pregnancies with cervical length (CL) ≤ 25 mm".

Line 109-186: In material and method section is not clear, rewrite again kindly, for data analysis which software are authors use?

We rephrased some sentences in order to make It clear. For data analysis we used the software “R” form R Foundation for Statistical Computing.

Line 159-167: Don’t use figure in methods and where is your statical analysis? and check grammar. 

Thank you for your guidance. This study used ultrasonography to obtain measurements of the cervix, and in addition there are different techniques to measure the cervix. Therefore, we believe this is a clear way of exemplifying how we obtained the measurements. Other references used figure in methods: 

Heath VC, Southall TR, Souka AP, Novakov A, Nicolaides KH. Cervical length at 23 weeks of gestation: relation to demographic characteristics and previous obstetric history. Ultrasound Obstet Gynecol. 1998;12(5):304-11.

The last two paragraphs of material and methods section describe statistical analyzes

 Line 189-191: Please add these lines in “material and method” section

I appreciate your comment. In this study, we used the STROBE checklist (STrengthening the Reporting of Observational studies in Epidemiology). In the results, item number 13, the description of the participants is oriented: (a) Report numbers of individuals at each stage of study—eg numbers potentially eligible, examined for eligibility, confirmed eligible, included in the study, completing follow-up, and analyzed.

Line 192: Don’t use figure, please rewrite this.

The STROBE suggests considering the possibility of use a flow diagram. We chose to use the flowchart to better illustrate eligible and excluded participants.

 Line 227-238: Didn’t need use any figure here.

I really consider your comments, however, as the main objective of this study was to construct curves with measurements of the cervix of pregnant women in the second trimester of pregnancy, we thought it would be interesting to present such curves.

Line 245: Check Grammar

Thank you for your comment. checked.

Line 277: Check Grammar

Thank you.

checked.

 Line 296: Check the spelling

We corrected the spelling: “shorter cervix”.

Line 327 – 332: The first sentence is not clear. It seems incomplete. Rewrite the full lines.

We rewrote the sentence:

“We found that CL was shorter in pregnant women ≤19 years old than those >20 years old. In the literature, young pregnant women are at increased risk for spontaneous PTB [30], which may be due to biological immaturity of the female genital tract [31,32], social and behavioral factors [18], and intra-amniotic infections as a consequence of genital tract infections [31]”.

Line 378-383: Rewrite the conclusion again

Thank you for your suggestion. We rewrote the conclusion.

“The reference CL distribution curves should consider populational characteristics since physicians may use it as a strategy to prevent preterm birth in clinical practice. Doing so, it will enable a more efficient diagnosis of short cervix and its association with prematurity, allowing assertive medical decisions. Moreover, we suggest that subsequent studies should consider these populational characteristics to build new distribution curves and define specific screening strategies for different populations to prevent premature delivery”.

---

## [Decision Letter · Decision Letter 1]

14 Jan 2022

PONE-D-21-13533R1Cervical length distribution among Brazilian pregnant population and risk factors for short cervix: a multicenter cross-sectional studyPLOS ONE

Dear Dr. Pacagnella,

Thank you for submitting your manuscript to PLOS ONE. After careful consideration, we feel that it has merit but does not fully meet PLOS ONE’s publication criteria as it currently stands. Therefore, we invite you to submit a revised version of the manuscript that addresses the points raised during the review process.

We look forward to receiving your revised manuscript.

Kind regards,

Federico Ferrari

Academic Editor

PLOS ONE

Additional Editor Comments:

Please follow the comments of the reviewers.

Reviewers' comments:

Reviewer's Responses to Questions

**Comments to the Author**

1. If the authors have adequately addressed your comments raised in a previous round of review and you feel that this manuscript is now acceptable for publication, you may indicate that here to bypass the “Comments to the Author” section, enter your conflict of interest statement in the “Confidential to Editor” section, and submit your "Accept" recommendation.

Reviewer #1: All comments have been addressed

Reviewer #2: All comments have been addressed

2. Is the manuscript technically sound, and do the data support the conclusions?

Reviewer #1: Partly

Reviewer #2: Yes

3. Has the statistical analysis been performed appropriately and rigorously? 

Reviewer #1: Yes

Reviewer #2: Yes

4. Have the authors made all data underlying the findings in their manuscript fully available?

Reviewer #1: Yes

Reviewer #2: Yes

5. Is the manuscript presented in an intelligible fashion and written in standard English?

Reviewer #1: Yes

Reviewer #2: Yes

6. Review Comments to the Author

Reviewer #1: Review of Revision

Abstract

Mention use of both techniques in objective

Consider including that it's a secondary analysis here

Are these stats adjusted OR or just OR? If adjusted I would change "OR" to "aOR".

Conclusion: I would consider mentioning more specifically what the study found after aOR: Something like: "Lower BMI and prior miscarriage or preterm birth are associated with CL <25mm"

Intro

Line 82 - word "birth" is missing after preterm

Line 90 - "Obstetricians and Gynecologists"

Methods

What is the power calculation for exactly? rate of CLs <25 mm? preterm birth?

What was the primary outcome?

I would clarify that there are two separate analyses being described. the two groups are separated by CL <25 and then the use of the different CL measurements.

This section needs to be clearer exactly what was done.

Results

Overall this section is confusing. I think getting rid of some of the text and referring more to tables might help. I think stressing the important findings and leaving the details to the tables would help streamline this section as well. It is hard to identify what the focus of the paper truly is.

Line 222 - not sure what this means

Fig 4 - why no means or percentages of those <25 mm in the last column for curved measurements?

Tables 3/4/5 - Should these say "multivariable analysis"? if yes, then the text should be changed from "OR" to "aOR"

Discussion

Line 291 - Not sure what point they are making regarding the 25th %ile. Is this a higher risk group in your cohort? How is 32mm different from 25th %ile previously described?

Line 302 - How were they different? Why might that be (based on population studied, inclusion/exclusion criteria etc)?

The discussion regarding straight vs curved could be shortened and combined.

Line 362 - is there data linking this to preterm birth?

Limitations should include exclusion of those with cerclage and who are dilated as this eliminates a large chunk of short cervix patients (this may also explain why your CL %iles might be slightly different in your population if other studies included those subjects)

Reviewer #2: In this manuscript, Marquat and collaborators present results of the association between cervical length distribution and risk factors for short cervix. The study was conducted with 8,167 pregnant women in Brazil. The authors should correct some simple errors before publication.

• Line 47: Have spelling in this word “secund”, it will be ‘second’.

• Line 90: Use just one space before ‘including’ word; It will be ‘American College of Obstetrics and Gynecology (ACOG)’.

• Line 195: Please rewrite this sentence as like “Statistical analysis is performed using R software from the R Project for Statistical Computing (version *.*.0)”.

• Line 254: Please check grammar.

• Line 286: Please check grammar.

• Line 343: Please rewrite “straight-line”.

• Line 363: Use just one space.

• Line 365: Use just one space.

• Line 414: Rewrite “birth in”.

7. PLOS authors have the option to publish the peer review history of their article (what does this mean?). If published, this will include your full peer review and any attached files.

Reviewer #1: **Yes: **Rachel Harrison

Reviewer #2: **Yes: **Dil Ware Alam

---

## [Author Response · Author response to Decision Letter 1]

3 Jun 2022

Reviewer 1

Intro

Line 82 - word "birth" is missing after preterm

R: Thank you for the suggestion. We rewrote: “TVU can also help to prevent prematurity because cervical length is one of the best predictors of preterm birth (PTB)”

Line 90 - "Obstetricians and Gynecologists"

R: Checked.

Methods

What is the power calculation for exactly? rate of CLs <25 mm? preterm birth?

R: The objective is to know the profile of measurements of the cervical length of Brazilian pregnant women in the second trimester of pregnancy and, consequently, which variables are related to measurements of the cervix ≤ 25 mm. We know that cervical measurements ≤ 25 mm are related to an increased risk for preterm birth.

What was the primary outcome?

R: The primary outcome is the construction of a distribution curve for cervical length measurements of Brazilian pregnant women in the second trimester of pregnancy (between 18 to 22 weeks and 6 days) using ultrasound (straight technique).

I would clarify that there are two separate analyses being described. the two groups are separated by CL <25 and then the use of the different CL measurements.

This section needs to be clearer exactly what was done.

R: We appreciate your suggestion. However, there are no two separate groups. We have a population of 7844 pregnant women in the second trimester of pregnancy. Straight-line cervical length measurements were obtained in all of them (standard measurement) and from then on, the percentage of short cervix per variable (maternal age, schooling, marital status, region, race, BMI, obstetric and gynecological history and gestational age...) was observed. In a secondary analysis, we compared the standard technique with cervical length using the curved technique and the volume of the cervix.

We rewrote it to make it cleareR: 

“Four strategies of uterine cervical measurements were used in our study: straight line measurement (SL) between the internal to the external os, used for the primary outcome (distribution); curved measurement (CM) with two straight measurements respecting the endocervical canal pathway between the internal and external os (Fig 1 and 2); anteroposterior measurement near the insertion of the uterine arteries, in the middle third of the cervix; and transverse measurement rotating the transducer 90 degrees to allow transverse visualization of the cervix. The volume of the cervix was calculated using the formula for the volume of a cylinder, πR²h, where R is half the transverse diameter of the cervix, and h is the length. The curved measurement and the measurements for calculating the volume were used only for comparison purposes with the standard straight measurement.”

Results

Overall this section is confusing. I think getting rid of some of the text and referring more to tables might help. I think stressing the important findings and leaving the details to the tables would help streamline this section as well. It is hard to identify what the focus of the paper truly is.

R: Thank you for the suggestion. We reduced some of the information repeated in tables. 

Line 222 - not sure what this means

R: As described in other studies and initially noted by Iams et al., pregnant women with a cervical length ≤25 mm are at increased risk of preterm birth than those with a cervical length > 25 mm. As we calculated the percentile, we are reporting our result that in our population, women with a percentile below the 5th have a cervix ≤25 mm and, consequently, higher risk of preterm birth.

Fig 4 - why no means or percentages of those <25 mm in the last column for curved measurements?

R: Thank you for highlighting this. We completed the missing values in table 2

Tables 3/4/5 - Should these say "multivariable analysis"? if yes, then the text should be changed from "OR" to "aOR"

R: Indeed. We changed in text.

Discussion

Line 291 - Not sure what point they are making regarding the 25th %ile. Is this a higher risk group in your cohort? How is 32mm different from 25th %ile previously described?

R: When we compared the lengths of the uterine cervix for the 5th, 10th and 25th percentiles, respectively, we observed values similar to the Iams study. Some studies consider that pregnant women with a uterine cervix measurement <30 mm are at increased risk for preterm delivery, and this value corresponds to the 25th percentile of the Iams study. In our population, pregnant women in the 25th percentile correspond to 32 mm, and therefore, considering the 25th percentile as an increased risk for premature birth, patients with measurement < 32 mm would be at increased risk. We understand that this information may be causing confusion and as such we have chosen to remove it.

Line 302 - How were they different? Why might that be (based on population studied, inclusion/exclusion criteria etc)?

R: We changed the paragraph to adapt it to the suggestion.

The discussion regarding straight vs curved could be shortened and combined.

R: We reduced the discussion to fit in two short paragraphs containing important information regarding the issue. 

Line 362 - is there data linking this to preterm birth?

Limitations should include exclusion of those with cerclage and who are dilated as this eliminates a large chunk of short cervix patients (this may also explain why your CL %iles might be slightly different in your population if other studies included those subjects)

R: Patients under cervical stiches are rare in Brazil and we understand that this would not have influenced int the distribution of cervical length.

Reviewer 2

Line 47: Have spelling in this word “secund”, it will be ‘second’.

R: Thank you. We correct.

Line 90: Use just one space before ‘including’ word; It will be ‘American College of Obstetrics and Gynecology (ACOG)’

R: Thank you very much. We correct.

Line 195: Please rewrite this sentence as like “Statistical analysis is performed using R software from the R Project for Statistical Computing (version *.*.0)”.

R: We rewrote the sentence. 

Line 254: Please check grammar.

R: Thank you. I believe you are referring to the "were" in this sentence: The variables and percentage of CL ≤25 mm were as follows”. However, as the context is in the plural (variables and percentage), that's why "were" is being used instead of "was". 

Line 286: Please check grammar

R: Checked. We use “were” for the plural (the risk factors were). “Risk factors associated with increased risk of CL ≤25 mm were as follows”

Line 343: Please rewrite “straight-line”.

R: Thank you. We rewrote.

 Line 363: Use just one space

R: Checked.

Line 365: Use just one space

R: Checked.

 Line 414: Rewrite “birth in”

R: Thank you. We rewrote.

---

## [Decision Letter · Decision Letter 2]

14 Jul 2022

Cervical length distribution among Brazilian pregnant population and risk factors for short cervix : a multicenter cross-sectional study

PONE-D-21-13533R2

Dear Dr. Pacagnella,

We’re pleased to inform you that your manuscript has been judged scientifically suitable for publication and will be formally accepted for publication once it meets all outstanding technical requirements.

Kind regards,

Federico Ferrari

Academic Editor

PLOS ONE

Additional Editor Comments (optional):

Reviewers' comments:

Reviewer's Responses to Questions

**Comments to the Author**

1. If the authors have adequately addressed your comments raised in a previous round of review and you feel that this manuscript is now acceptable for publication, you may indicate that here to bypass the “Comments to the Author” section, enter your conflict of interest statement in the “Confidential to Editor” section, and submit your "Accept" recommendation.

Reviewer #2: All comments have been addressed

2. Is the manuscript technically sound, and do the data support the conclusions?

Reviewer #2: Yes

3. Has the statistical analysis been performed appropriately and rigorously? 

Reviewer #2: Yes

4. Have the authors made all data underlying the findings in their manuscript fully available?

Reviewer #2: Yes

5. Is the manuscript presented in an intelligible fashion and written in standard English?

Reviewer #2: Yes

6. Review Comments to the Author

Reviewer #2: (No Response)

7. PLOS authors have the option to publish the peer review history of their article (what does this mean?). If published, this will include your full peer review and any attached files.

Reviewer #2: **Yes: **DIL WARE ALAM

---

## [Editor Report · Acceptance letter]

28 Sep 2022

PONE-D-21-13533R2 

Cervical length distribution among Brazilian pregnant population and risk factors for short cervix: a multicenter cross-sectional study 

Dear Dr. Pacagnella:

I'm pleased to inform you that your manuscript has been deemed suitable for publication in PLOS ONE. Congratulations! Your manuscript is now with our production department. 

Kind regards, 

on behalf of

Dr Federico Ferrari 

Academic Editor

PLOS ONE